# Prediction of differentiation levels in lung adenocarcinoma using peripheral blood inflammatory cytokines and tumor markers

Yang Li[1], Jiahuan Wu[1], Meiling Long[1], Tingting Zeng[2], Depeng Jiang[1]*

**1** Department of Respiratory Medicine, The Second affiliated Hospital of Chongqing Medical University, Chongqing, China, **2** Department of Endocrinology, The Second affiliated Hospital of Chongqing Medical University, Chongqing, China

☯ These authors share first authorship

* gdp116@hospital.cqmu.edu.cn

## Abstract

### Objective

Lung Adenocarcinoma (LUAD) has highly aggressive and lethal, and its degree of differentiation significantly influences prognosis and treatment strategies, yet accurate prediction remains challenging. To assess the predictive value of combining peripheral blood inflammatory markers, such as the aggregate index of systemic inflammation (AISI), with tumor markers, including Carcinoembryonic Antigen (CEA) and Cytokeratin 19 fragment antigen 21−1(CYFRA21−1), etc, for determining LUAD differentiation levels.

### Methods

This retrospective study included 203 LUAD patients treated at Chongqing Medical University's Second Affiliated Hospital, categorized by low and high differentiation. Demographic, clinical, and laboratory data including peripheral blood inflammatory and tumor markers were analyzed. A multivariate logistic regression model evaluated these markers' predictive accuracy.

### Results

AISI (OR = 1.64, 95% CI = 1.08–2.58, $p$ = 0.024), CEA (OR = 1.02, 95% CI = 1.00–1.04, $p$ = 0.0497), ferritin (OR = 1.01, 95% CI = 1.00–1.01, $p$ = 0.010), and Progastrin Releasing Peptide (ProGRP) (OR = 1.03, 95% CI = 1.00–1.07, $p$ = 0.047) were risk factors of low differentiation LUAD. The model achieved an Area Under Curve(AUC) of 0.795 (95%CI: 0.726–0.864) for distinguishing low from high differentiation, with decision curve analysis confirming clinical utility.

**Data availability statement:** The data used to support the findings of this study are available from the corresponding author upon reasonable request. If you need to further verify the authenticity of the data, you can contact the Ethics and Morality Committee via email (chongerlcyj@163.com).

**Funding:** This study was funded by Chongqing Natural Science Foundation (CSTB2022NSCQ-MSX0127).

**Competing interests:** The authors have declared that no competing interests exist.

## Conclusion

This model, combining inflammatory and tumor markers, effectively predicts LUAD differentiation, aiding personalized treatment planning, enhancing therapeutic outcomes, and supporting early LUAD detection.

## Introduction

Lung cancer is a leading cause of cancer incidence and mortality globally, posing a serious threat to human health. According to global cancer statistics (GLOBOCAN), approximately 2.2 million new lung cancer cases and 1.8 million related deaths occurred in 2020 [1]. Lung cancer is broadly divided into small cell lung cancer and non-small cell lung cancer (NSCLC), with NSCLC making up over 85% of cases. Lung Adenocarcinoma (LUAD), the most prevalent NSCLC subtype, accounts for around 40% of cases, followed by squamous cell carcinoma at 25%. Despite advances in diagnosis and treatment, LUAD remains highly aggressive and lethal, with a five-year survival rate below 5% [2]. Factors such as tumor size, differentiation level, Tumor Node Metastasis classification(TNM) stage, and lymph node metastasis significantly influence LUAD treatment outcomes and survival.

The connection between inflammation and cancer has become a primary focus in recent research. Inflammatory responses are crucial for tissue repair after tumor-induced damage and can influence tumor microenvironment stability, exhibiting both tumor-suppressive and tumor-promoting effects [3]. Studies indicate that systemic inflammatory markers, such as neutrophil-to-lymphocyte ratio (NLR) and platelet-to-lymphocyte ratio (PLR), are closely linked with tumor invasion, recurrence, metastasis, and prognosis [4–5]. These ratios are key indicators of systemic inflammation and strongly correlate with cancer outcomes [6–7]. Given LUAD's high mortality and complex pathology, identifying precise diagnostic and prognostic tools is essential [8]. Tumor differentiation plays a central role in assessing treatment response and survival, requiring accurate evaluation to guide personalized therapy. However, traditional histopathology faces limitations like subjectivity and delayed results in evaluating tumor differentiation [9].

This study evaluated the clinical utility of combining peripheral blood inflammatory markers aggregate index of systemic inflammation (AISI) with specific tumor markers to predict differentiation levels in LUAD. We hypothesized that particular marker combinations may more accurately reflect the biological characteristics of LUAD, offering clinicians a novel, non-invasive tool for assessing tumor differentiation. This method could improve treatment decisions by enhancing specificity and therapeutic effectiveness.

## Methods and materials

### Study population

This retrospective study included 203 lung adenocarcinoma patients consecutively admitted to the Second Affiliated Hospital of Chongqing Medical University from June 1,

2022, to July 30, 2024. Data were accessed between December 1, 2024 and January 1, 2025. This study utilized anonymized medical records from the Second Affiliated Hospital of Chongqing Medical University. Inclusion criteria were as follows: age 18–90 years, LUAD was initially confirmed by pathological diagnosis and complete clinical and follow-up data. Exclusion criteria included: age < 18 or >90 years, lack of pathological evidence for malignant lung tumors, non-adenocarcinoma lung cancers, prior anti-tumor treatment, undefined pathological stage or differentiation level, and incomplete clinical or follow-up data. The study focused on patients with low and high differentiation levels, so cases with other differentiation levels were excluded. This resulted in two groups: low differentiation and high differentiation. The patient selection process is shown in Fig 1 [10]. All personal identifiers were removed prior to researcher access, and this retrospective study was conducted in accordance with the ethical principles of the Declaration of Helsinki and was approved by the Second Affiliated Hospital of Chongqing Medical University Ethics Committee (No.85, 2024), with a waiver of informed consent obtained.

## Definition of low and high differentiation

Low differentiated LUAD refers to cancer cells with minimal differentiation, showing marked deviations from the epithelial cells of normal lung glands. These cells exhibit irregular shapes, vary in size, have enlarged and darkly stained nuclei, and feature an increased nucleus-to-cytoplasm ratio, indicating a loss of normal cellular characteristics (Figure SIa). High differentiated LUAD consists of cancer cells that are highly differentiated, closely resembling normal lung gland epithelial cells in morphology. These cells appear regular, with smaller nuclei and a normal nucleus-to-cytoplasm ratio (Figure SIb) [11].

## AISI calculation and pathological differentiation definition

The AISI score was calculated using the formula: neutrophils×platelets× monocytes/lymphocytes [6].

## Statistics analysis

For the examination of categorical variables, the chi-square test or Fisher's exact test was chosen as the preferred method. To assess continuous variables, both the Mann-Whitney U test and the independent samples T test were employed. Due to the binary nature of our dependent variable, logistic regression models were employed. Multivariate binary logistic regression models were constructed based on univariate logistic regression results. The performance of

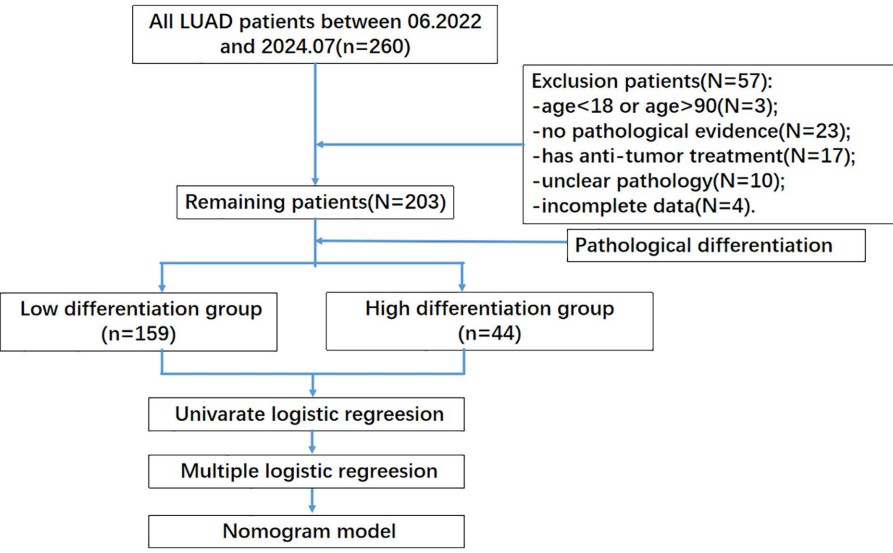

**Fig 1. The procedure of the study.**

the model was evaluated through discrimination, as measured by the Area Under the Receiver Operating Characteristic Curve (ROC). Clinical utility was evaluated using decision curve analysis (DCA), and internal validation was conducted via bootstrapping with 1,000 repetitions. Statistical analyses were performed using R version 4.4.3, with P-value < 0.05 considered statistically significant.

## Results

### Baseline characteristics of study population

The study retrospectively analyzed 203 patients with lung adenocarcinoma, including 109 men (53.69%) and 94 women (46.31%). Most patients did not report smoking or drinking habits, and a minority had chronic conditions such as diabetes (18 patients, 8.87%), hypertension (54 patients, 26.6%), and coronary heart disease (10 patients, 4.93%). All cases were pathologically confirmed as lung adenocarcinoma and classified into stages 0–4 according to American Joint Committee on Cancer Tumor-Node-Metastasis Staging System(AJCC TNM Staging). Based on pathologist evaluations, patients were categorized into low-differentiation (159 patients, 78.33%) and high-differentiation (44 patients, 21.67%) groups. Table 1 presents the detailed baseline characteristics of the cohort.

**Table 1. Baseline characteristics.**

| Variables | Analysis set (n = 203) |
| --- | --- |
| Sex,n(%) | |
| Male sex | 109 (53.69%) |
| Female sex | 94 (46.31%) |
| Smoke,n(%) | |
| No | 126 (62.07%) |
| Yes | 77 (37.93%) |
| Drink,n(%) | |
| No | 149 (73.40%) |
| Yes | 54 (26.60%) |
| Diabetes,n(%) | |
| No | 185 (91.13%) |
| Yes | 18 (8.87%) |
| Hypertension,n(%) | |
| No | 149 (73.40%) |
| Yes | 54 (26.60%) |
| CAD,n(%) | |
| No | 193 (95.07%) |
| Yes | 10 (4.93%) |
| AJCC TNM | |
| 0 | 2 (0.98%) |
| 1 | 11 (5.42%) |
| 2 | 11 (5.42%) |
| 3 | 31 (15.27%) |
| 4 | 148 (72.91%) |
| Degree of differentiation | |
| Low | 159 (78.33%) |
| High | 44 (21.67%) |

AJCC TNM, AJCC Tumor-Node-Metastasis (TNM) Staging System;BMI, body mass index; CAD, coronary artery disease.

Table 2 presents a descriptive analysis of the demographic characteristics, routine peripheral blood test results, and standard lung cancer tumor markers for the study population.The average age was 62.10±11.54 years, and the mean body mass index (BMI) was 22.96±3.40. The mean AISI was 5.81±1.03, with a median of 5.77. Median values for tumor markers were as follows: Carcinoembryonic Antigen (CEA) at 7.16, Cytokeratin 19 fragment antigen 21−1(CYFRA21−1) at 2.97, ferritin at 143.06, Neuron Specific Enolase (NSE) at 13.86, Squamous Cell Carcinoma Antigen (SCCA) at 0.70, Tissue Polypeptide Antigen (TPA) at 103.6, and Progastrin Releasing Peptide (ProGRP) at 31.68.

## Analysis of LUAD patients with different differentiation levels

In this study, LUAD differentiation was classified by pathologists based on the resemblance between tumor cells and normal lung tissue cells, dividing patients into low differentiation (159 patients, 78.33%) and high differentiation groups (44 patients, 21.67%). Findings are summarized in Table 3. Baseline characteristics showed no statistically significant differences between the groups (P>0.05). However, significant differences were observed in laboratory test results, including neutrophils (P=0.001), monocytes (P=0.002), platelets (P=0.008), ln-AISI (P=0.002), CEA (P<0.001), CYFRA21−1 (P<0.001), ferritin (P<0.001), NSE (P=0.016), TPA (P<0.001), and ProGRP (P=0.004). Due to the limited number of low-differentiated patients in stages 0–2 (only one case), statistical analysis was not feasible for this subgroup. Therefore, the primary analysis focused on stages 3–4, with detailed results presented in the supplementary materials and baseline characteristics shown in S1 Table in S1 File.

Due to skewed laboratory test data, the Mann-Whitney U test was applied (Table 4), revealing statistically significant differences in CEA, CYFRA21−1, and TPA values (p<0.05).The skewed laboratory test data for the stage 3–4 group is presented in Table S2 in S1 Table.

The multivariate logistic regression model assessed the probability of low differentiation in lung adenocarcinoma patients (Table 5). Key influencing factors were identified through a backward stepwise regression analysis, and the results are presented in Fig 2. Key predictors included AISI (OR = 1.64, 95% CI = 1.08–2.58, p=0.024), CEA (OR = 1.02, 95% CI = 1.00–1.04,

**Table 2. Analysis of the demographic characteristics.**

| Variables | Min | P25 | P50 | Mean±sd | P75 | Max |
|---|---|---|---|---|---|---|
| Age(years) | 26 | 55 | 64 | 62.10±11.54 | 72 | 87 |
| Height(cm) | 145 | 155 | 160 | 160.94±7.47 | 168 | 178 |
| Weight(kg) | 40 | 52 | 60 | 59.45±9.63 | 65 | 85 |
| BMI(kg/m2) | 15.42 | 20.57 | 22.76 | 22.96±3.40 | 24.97 | 31.25 |
| Laboratory examination | | | | | | |
| Neutrophil | 1.59 | 3.37 | 4.42 | 5.00±2.99 | 6.18 | 36.46 |
| Lymphocyte | 0.52 | 0.98 | 1.33 | 1.39±0.53 | 1.73 | 4.64 |
| Monocyte | 0.16 | 0.33 | 0.46 | 0.49±0.23 | 0.58 | 2.16 |
| Platelet | 19 | 175 | 223 | 234.56±89.84 | 278 | 882 |
| lnAISI | 3.91 | 5.16 | 5.77 | 5.81±1.03 | 6.48 | 9.1 |
| CEA# | 0.2 | 2.79 | 7.16 | 67.78±178.77 | 35.52 | 1003 |
| CYFRA21−1# | 0.45 | 1.66 | 2.97 | 6.39±8.94 | 6.4 | 54.54 |
| Ferritin | 3.6 | 84.7 | 143.06 | 191.97±190.34 | 236.2 | 1500 |
| NSE | 8.34 | 11.79 | 13.86 | 16.21±10.63 | 16.62 | 122.3 |
| SCCA# | 0.1 | 0.5 | 0.7 | 1.09±1.79 | 1.1 | 21.5 |
| TPA# | 11.13 | 55.21 | 103.6 | 198.62±272.04 | 194.2 | 1783 |
| ProGRP | 8.55 | 23.04 | 31.68 | 37.98±37.38 | 41.54 | 427.69 |

# Nonnormal data, mean±standard deviation of non-normal data expressed as mean.

AISI: aggregate index of systemic inflammation; CEA: Carcinoembryonic Antigen; CYFRA21−1: Cytokeratin 19 fragment antigen 21−1; NSE: Neuron Specific Enolase; SCCA: Squamous Cell Carcinoma Antigen; TPA: Tissue Polypeptide Antigen; ProGRP: Progastrin Releasing Peptide.

**Table 3. Baseline characteristics and laboratory examinations of two groups.**

| Variables | Low group (n = 159) | High group (n = 44) | P-value |
|---|---|---|---|
| Sex,n(%) | | | 0.966 |
| Male sex | 86 (54.09%) | 23 (52.27%) | |
| Female sex | 73 (45.91%) | 21 (47.73%) | |
| Age(years) | 61.72 ± 11.77 | 63.48 ± 10.65 | 0.348 |
| Height(cm) | 161.12 ± 7.58 | 160.30 ± 7.13 | 0.505 |
| Weight(kg) | 59.43 ± 9.59 | 59.55 ± 9.87 | 0.944 |
| BMI(kg/m2) | 22.89 ± 3.35 | 23.20 ± 3.61 | 0.618 |
| Smoke,n(%) | | | 0.442 |
| No | 96 (60.38%) | 30 (68.18%) | |
| Yes | 63 (39.62%) | 14 (31.82%) | |
| Drink,n(%) | | | 1.000 |
| No | 117 (73.58%) | 32 (72.73%) | |
| Yes | 42 (26.42%) | 12 (27.27%) | |
| Diabetes,n(%) | | | 0.373 |
| No | 143 (89.94%) | 42 (95.45%) | |
| Yes | 16 (10.06%) | 2 (4.55%) | |
| Hypertension,n(%) | | | 0.489 |
| No | 119 (74.84%) | 30 (68.18%) | |
| Yes | 40 (25.16%) | 14 (31.82%) | |
| CAD,n(%) | | | 0.454 |
| No | 152 (95.60%) | 41 (93.18%) | |
| Yes | 7 (4.40%) | 3 (6.82%) | |
| Laboratory examination | | | |
| neutrophil | 5.26 ± 3.26 | 4.09 ± 1.43 | 0.001 |
| lymphocyte | 1.40 ± 0.56 | 1.34 ± 0.41 | 0.458 |
| monocyte | 0.51 ± 0.25 | 0.42 ± 0.14 | 0.002 |
| platelet | 241.61 ± 95.09 | 202.05 ± 50.06 | 0.008 |
| ln-AISI | 5.93 ± 0.94 | 5.45 ± 0.78 | 0.002 |
| CEA# | 83.53 ± 198.71 | 10.86 ± 28.62 | <0.001 |
| CYFRA21−1# | 7.34 ± 9.62 | 2.98 ± 4.52 | <0.001 |
| Ferritin | 210.10 ± 205.74 | 126.45 ± 95.08 | <0.001 |
| NSE | 16.80 ± 11.77 | 14.10 ± 4.00 | 0.016 |
| SCCA# | 1.14 ± 1.99 | 0.91 ± 0.69 | 0.222 |
| TPA# | 226.55 ± 296.91 | 97.68 ± 102.02 | <0.001 |
| ProGRP | 40.50 ± 41.20 | 28.88 ± 14.75 | 0.004 |

# Nonnormal data, mean ± standard deviation of non-normal data expressed as mean;

BMI, body mass index; CAD, coronary artery disease; AISI: aggregate index of systemic inflammation; CEA: Carcinoembryonic Antigen; CYFRA21−1: Cytokeratin 19 fragment antigen 21−1; NSE: Neuron Specific Enolase; SCCA: Squamous Cell Carcinoma Antigen; TPA: Tissue Polypeptide Antigen; ProGRP: Progastrin Releasing Peptide.

**Table 4. Nonnormal data of two groups.**

| Variables | Low group (n = 159) | High group (n = 44) | P-value |
|---|---|---|---|
| CEA# | 114.66 | 56.26 | <0.001 |
| CYFRA21−1# | 112.75 | 63.16 | <0.001 |
| SCCA# | 104.14 | 94.27 | 0.322 |
| TPA# | 110.95 | 69.66 | <0.001 |

# Nonnormal data, mean ± standard deviation of non-normal data expressed as mean.

CEA: Carcinoembryonic Antigen; CYFRA21−1: Cytokeratin 19 fragment antigen 21−1; SCCA: Squamous Cell Carcinoma Antigen; TPA: Tissue Polypeptide Antigen.

**Table 5. The multiple logistic regression model.**

| Variable | OR(95%CI) | P-value |
|---|---|---|
| lnAISI | 1.76(1.04, 3.00) | 0.037 |
| Sex(male) | 0.41(0.13, 1.24) | 0.113 |
| Age | 0.96 (0.92, 1.00) | 0.036 |
| BMI | 1.01 (0.90, 1.14) | 0.855 |
| Somking | 1.38 (0.41, 4.61) | 0.604 |
| Drinking | 0.72 (0.24, 2.15) | 0.553 |
| CEA | 1.02 (1.00, 1.04) | 0.025 |
| CYFRA21−1 | 1.14 (0.81,1.59) | 0.459 |
| Ferritin | 1.01 (1.00,1.01) | 0.003 |
| NSE | 0.96 (0.88,1.04) | 0.321 |
| TPA | 1.00 (0.99,1.01 | 0.747 |
| ProGRP | 1.04(1.00,1.08) | 0.029 |

AISI: aggregate index of systemic inflammation; BMI, body mass index; CEA: Carcinoembryonic Antigen; CYFRA21−1: Cytokeratin 19 fragment antigen 21−1; NSE: Neuron Specific Enolase; SCCA: Squamous Cell Carcinoma Antigen; TPA: Tissue Polypeptide Antigen; ProGRP: Progastrin Releasing Peptide.

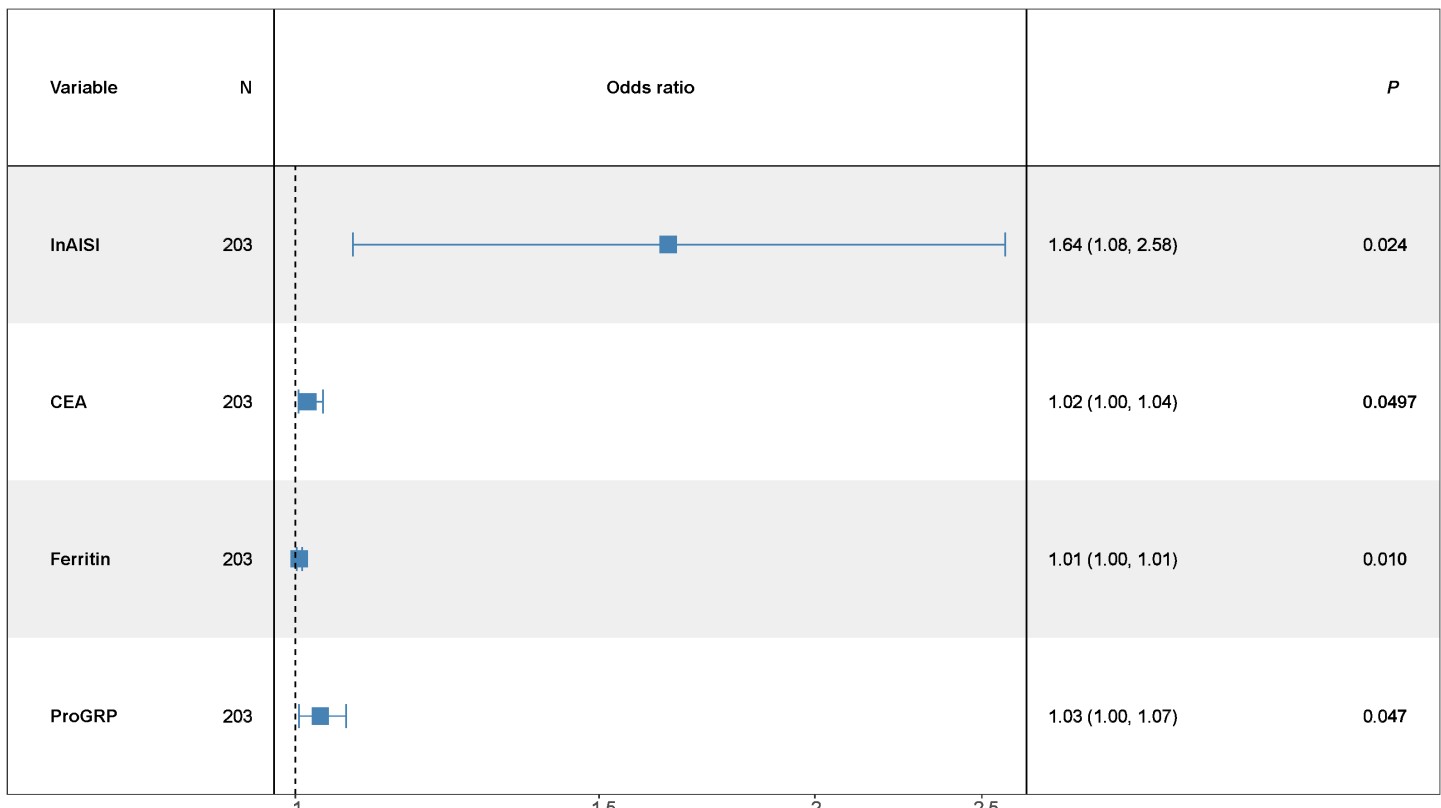

**Fig 2. The optimized multiple logistic regression model (backward stepwise regression).** (The odds ratio (OR) > 1.00, indicating that the variable is associated with an increased risk of poorly differentiated LUAD. AISI:Aggregate Index of Systemic Inflammation;CEA: Carcinoembryonic Antigen;Pro-GRP: Progastrin-Releasing Peptide).

p=0.0497), Ferritin (OR = 1.01, 95% CI = 1.00–1.01, p=0.010), and ProGRP (OR = 1.03, 95% CI = 1.00–1.07, p=0.047). To develop a nomogram for predicting the probability of low differentiation in lung cancer based on the optimized model (Fig 3), with no multicollinearity among the variables (Supplementary Table 3). The result showed that higher AISI, higher CEA, higher ferritin and higher ProGRP were risk factors. This nomogram offered a visual point – based system that is established on AISI, CEA, ferritin, and ProGRP to estimate the likelihood of poorly differentiated lung cancer. To calculate the probability of poorly differentiated lung cancer, the points assigned to each of the four variables, as determined on the scale, are summed up to obtain a total score. A vertical line is then drawn from the total point scale to the final axis, and the corresponding probability of poorly differentiated lung cancer can be obtained. The model achieved an AUC of 0.795 (95% CI: 0.726–0.864), as shown in Fig 4. The calibration curve was plotted using the Bootstrap method with 1,000 repetitions. The dashed diagonal line represents the ideal line, which signifies a perfect prediction. The solid line indicates the actual prediction performance of the model. The closer these two lines are to each other, the better the model's predictive accuracy. The results are presented in Fig 5. The absolute error between the simulated curve and the actual curve is 0.012, demonstrating that the two curves follow a highly similar trend and exhibit strong consistency. The decision curve for the established nomogram was constructed. The clinical value of this model increases as its curve moves away from the "treat all" or "treat none" curves. The threshold at which patients benefit from this model ranges from 30% to 95%. Furthermore, the decision curve analysis depicted in Fig 6 confirms the clinical utility of the model, demonstrating a favorable net benefit rate. This analysis further validates the model's effectiveness in clinical practice.

## Discussion

This study demonstrated that peripheral blood biomarkers and tumor biomarkers were closely associated with pathological differentiation in LUAD. By integrating these markers, we developed a predictive model with an AUC of 0.795 (95%CI:

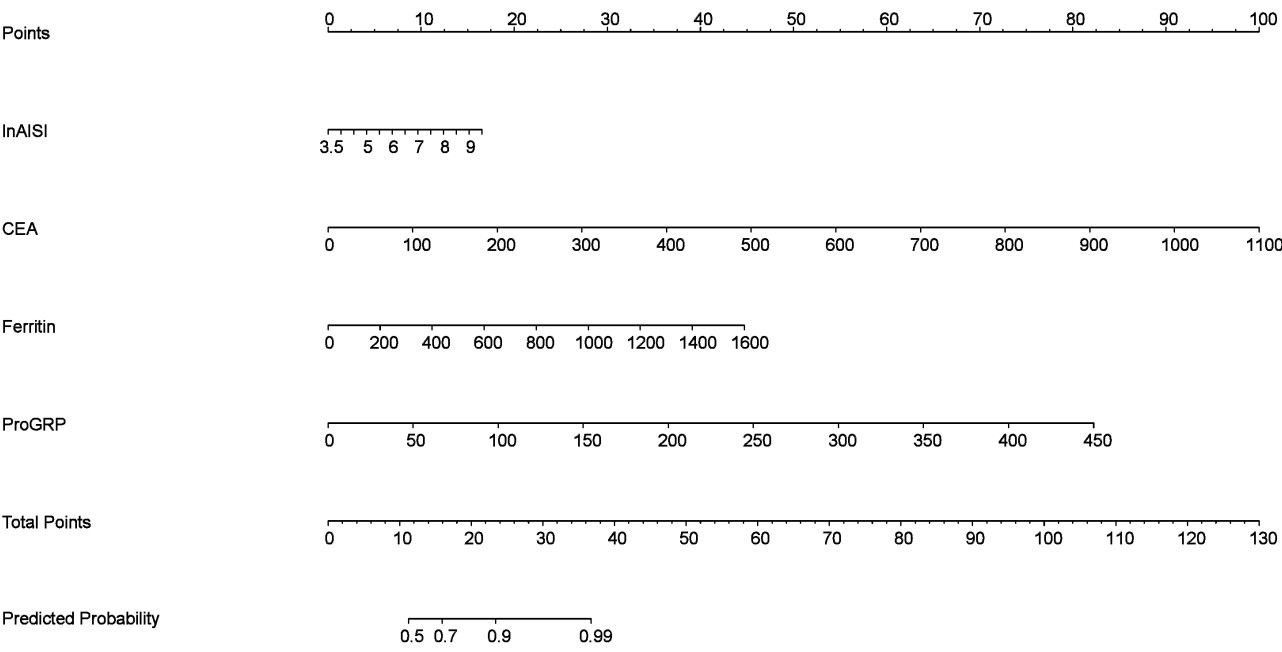

**Fig 3. The nomogram model to predict low differentiation in lung cancer.** (Using the nomogram, clinicians can derive a score based on individual patient characteristics. This total score is then translated into a probability of risk of poorly differentiated LUAD, facilitating stratified management and personalized interventions for higher risk patients. AISI: Aggregate Index of Systemic Inflammation; CEA: Carcinoembryonic Antigen; ProGRP: Progastrin-Releasing Peptide).

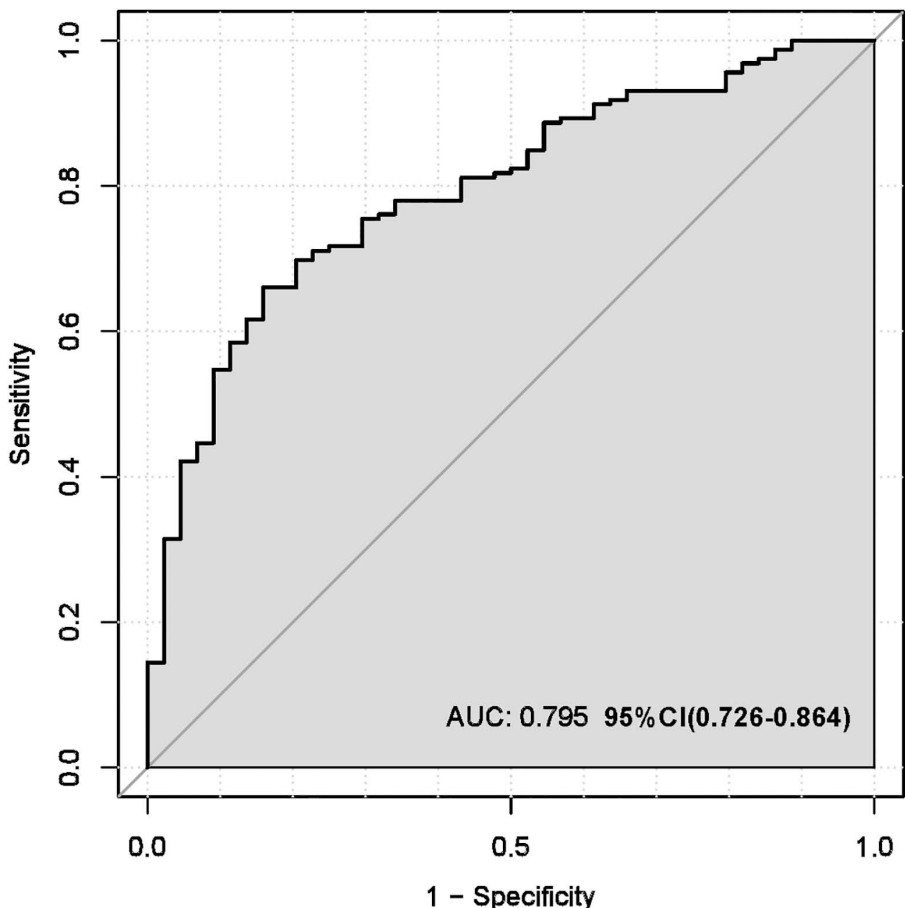

**Fig 4. The AUC of the nomogram model.** ( (AUC is 0.795 (95% CI: 0.726 - 0.864), indicateing that the model possesses a relatively good discriminatory ability and accuracy).

0.726–0.864), outperforming traditional indices such as SIRI (AUC:0.625) and LDH (AUC:0.596). The findings align with recent advances in biomarker-driven oncology, which emphasize the synergistic role of tumor biology and systemic inflammation in cancer progression [12–15].

The differentiation status of LUAD is a critical prognostic factor. Well differentiated tumors typically exhibit slow growth and localized spread, leading to higher patient survival rates and more favorable treatment outcomes. In contrast, poorly differentiated tumors are characterized by high aggressiveness, early metastasis, and heightened resistance to therapy [16–19]. Our model's ability to noninvasively predict differentiation addresses a clinical gap by reducing reliance on invasive biopsies, thereby improve detection rates and reduce healthcare costs. This aligns with NCCN guidelines advocating for non-invasive biomarker integration in NSCLC management [20–24]. However, as a pathological feature, the biological mechanism of differentiation degree may involve the dynamic interaction between tumor microenvironment (TME) and angiogenesis. The heterogeneity of TME may affect the differentiation status of LUAD [25]. Well differentiated LUAD usually presents with intact glandular structure and low nuclear atypia, while poorly differentiated LUAD is often accompanied by an increase in micropapillary/solid components, suggesting the difference between immune cell infiltration and matrix remodeling in the TME. The potential mechanisms are as follows: (1) Inflammatory cell infiltration: Poorly differentiated LUAD may promote the formation of the

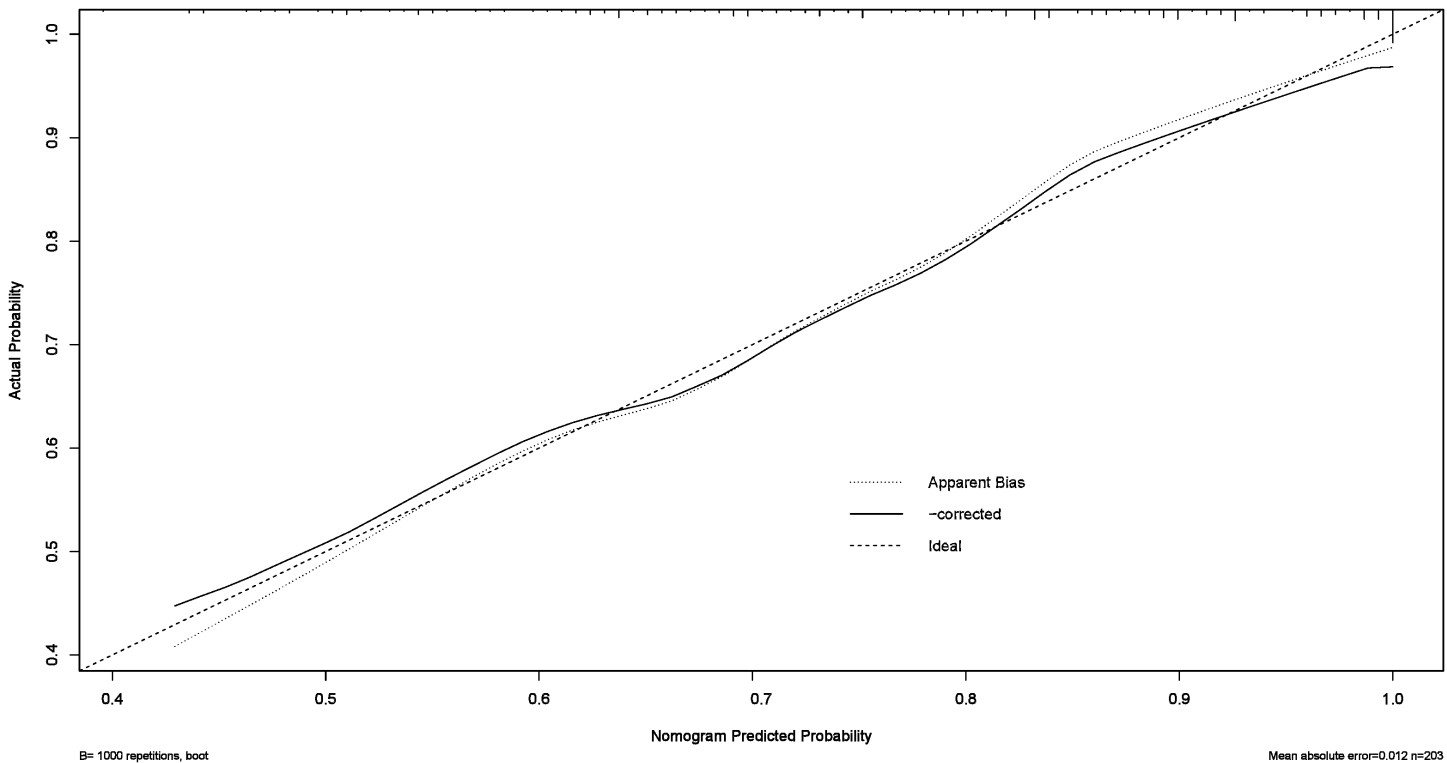

**Fig 5. The calibration curve of the nomogram model.** (The calibration curve was generated using 1,000 bootstrap repetitions. The diagonal dashed line represents the ideal case of perfect prediction, while the solid line indicates the actual performance of our model. Closer agreement between the two lines signifies better predictive accuracy).

immunosuppressive microenvironment by recruiting M2-type macrophages or regulatory T cells (Tregs), and inhibit tumor differentiation-related signaling pathways (such as Wnt/β-catenin) [26–27]. (2) Extracellular matrix (ECM) remodeling: In poorly differentiated tumors, fibroblast activation is enhanced, secreting large amounts of collagen and hyaluronic acid, which may interfere with the epithelial cell differentiation process through mechanical pressure or signaling molecules (such as TGF-β) [28]. (3) Metabolic reprogramming: Poorly differentiated LUAD may rely on the glycolytic pathway (Warburg effect), leading to lactic acid accumulation and inhibiting the reverse process of epithelial-mesenchymal transition (EMT), thereby maintaining an undifferentiated state [29]. Previous studies had shown that inflammatory markers such as AISI were associated with the prognosis of LUAD, but few studies on the differentiation mechanism.

Inflammatory markers, including AISI and neutrophils, reflect a tumor microenvironment that promotes angio-genesis and immune evasion. This is complemented by tumor biomarkers such as CEA and CYFRA 21–1, which directly quantify malignant cellular activity. This integrative approach provides a holistic view of LUAD behavior, consistent with studies highlighting the interplay between inflammation and tumor differentiation [21,30–32]. Notably, the elevated CYFRA21–1 levels observed in poorly differentiated LUAD may stem from increased cytokeratin-19 fragment release during rapid tumor proliferation, a phenomenon previously reported in undifferen-tiated malignancies [33–35].

Secondary markers such as ferritin and ProGRP offer additional insights into LUAD heterogeneity and improve model performance. Elevated ferritin suggests dysregulated iron metabolism, which can promote tumor

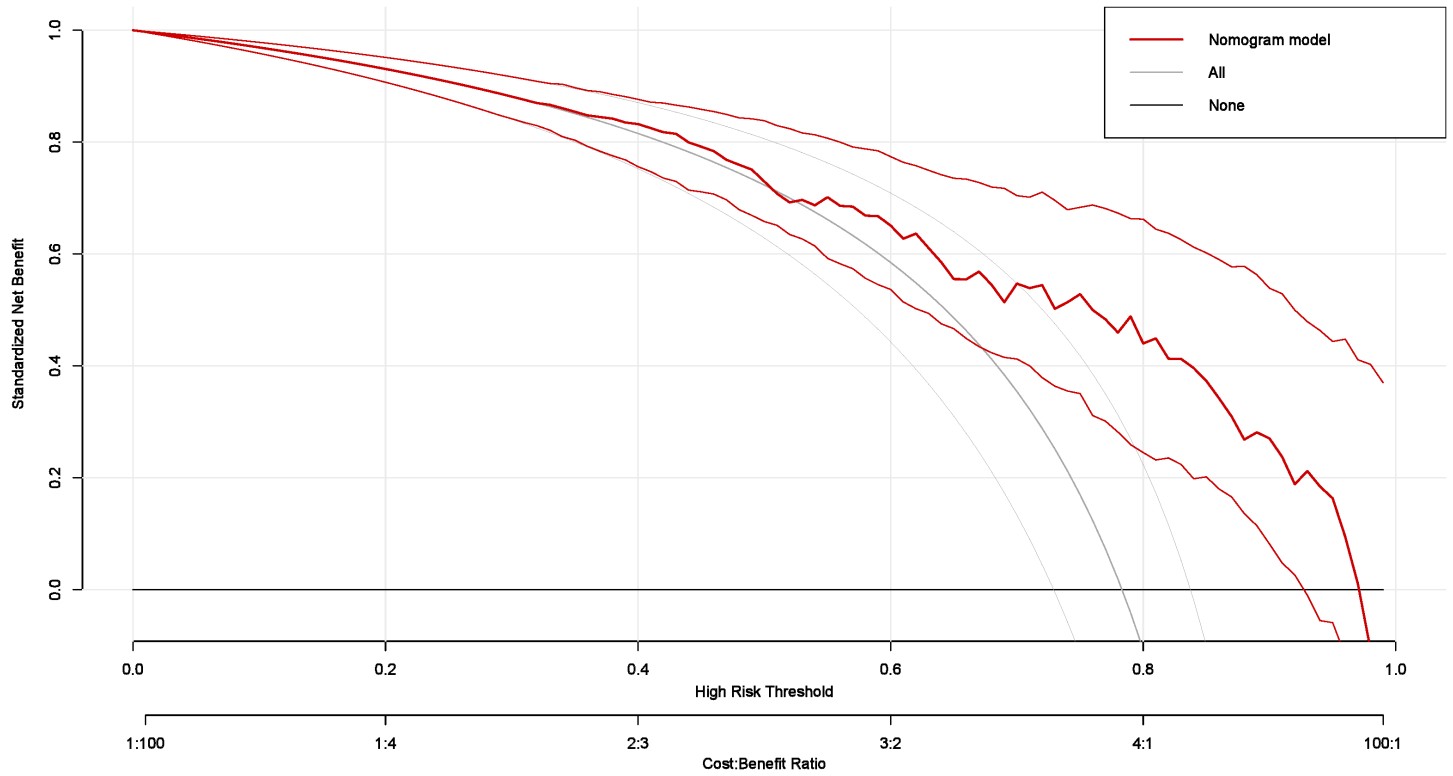

**Fig 6. The decision curve of the nomogram model.** (The nomogram offered a greater net benefit compared to both the "treat-all" and "treat-none" approaches over a threshold probability range from 30% to 95%, confirming its clinical utility in routine practice).

progression through oxidative stress and hypoxia adaptation [36–38]. Similarly, ProGRP detection in LUAD indicates an overlap with neuroendocrine mechanisms, warranting further investigation into its utility for subtype classification [39–41].

However, limitations must be acknowledged. First, although multicollinearity tests were employed during model construction to reduce the risk of overfitting, this risk remains in studies with limited sample sizes. Second, the retrospective data used in this study were solely sourced from the Second Affiliated Hospital of Chongqing Medical University, which may introduce potential geographic and population biases. We acknowledged that the model developed from this single center dataset may have limited generalizability, and its performance could decline when applied to populations from other regions, with different ethnic backgrounds, or under varying healthcare settings. Third, we recognized that a truly robust model requires validation with data from other independent centers, but the step had not yet been accomplished in this study. To directly address the limitations identified in this study, we propose several key directions for future research. First, multi center collaborations should be initiated to conduct rigorous external validation of our current model using independent datasets from various institutions. This is an essential step for confirming the model's clinical utility and generalizability. Second, large scale prospective cohort studies are warranted to elucidate the temporal relationship between AISI and the incidence and progression of lung adenocarcinoma. Third, integrating more precise diagnostic tools for lung adenocarcinoma differentiation could enhance the accuracy and reliability of research findings. Furthermore, developing predictive models that incorporate AISI with other biomarkers would help improve prognostic accuracy. Finally, fundamental experimental research focusing on

areas such as tumor microenvironment analysis and single-cell studies should be pursued to progressively address the current limitations of the research.

## Conclusion

This study suggested that inflammatory factors may serve as potential predictors for the differentiation of LUAD. By integrating inflammatory and tumor biomarkers, our predictive model effectively assessed pathological differentiation, aided in personalized treatment planning, and offered a reliable foundation for clinical decisions. Future studies should focued on elucidating the underlying biological mechanisms and validating the model's clinical efficacy.

## Supporting information

**S1 Fig. Supplementary figures.**
(DOCX)

**S1 Table. Supplementary tables.**
(DOCX)

## Acknowledgments

We thank the Second Affiliated Hospital of Chongqing Medical University for their support.

## Author contributions

**Conceptualization:** Yang Li, Meiling Long, DePeng Jiang.

**Data curation:** Yang Li, Meiling Long, Tingting Zeng.

**Formal analysis:** Yang Li.

**Investigation:** Yang Li.

**Methodology:** Yang Li, Tingting Zeng.

**Software:** Yang Li, Meiling Long.

**Supervision:** Jiahuan Wu.

**Validation:** Meiling Long, DePeng Jiang.

**Visualization:** Yang Li.

**Writing – original draft:** Yang Li, DePeng Jiang.

**Writing – review & editing:** Yang Li, Jiahuan Wu, DePeng Jiang.

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
