## [Decision Letter · Decision Letter 0]

14 Apr 2025

Dear Dr. Jiang,

Thank you for submitting your manuscript to PLOS ONE. After careful consideration, we feel that it has merit but does not fully meet PLOS ONE’s publication criteria as it currently stands. Therefore, we invite you to submit a revised version of the manuscript that addresses the points raised during the review process.

We look forward to receiving your revised manuscript.

Kind regards,

Zhiling Yu

Academic Editor

PLOS ONE

Additional Editor Comments:

1. Emphasize the need for prospective multi-center validation in the discussion and conclusion.

2. Expand the discussion to hypothesize pathways (e.g., tumor microenvironment interactions, angiogenesis).

3. Address minor grammatical errors (e.g., "improve the detection rate" → "improve detection rates").

Reviewers' comments:

Reviewer's Responses to Questions

**Comments to the Author**

1. Is the manuscript technically sound, and do the data support the conclusions?

Reviewer #1: Partly

Reviewer #2: Yes

2. Has the statistical analysis been performed appropriately and rigorously?

Reviewer #1: No

Reviewer #2: Yes

3. Have the authors made all data underlying the findings in their manuscript fully available?

Reviewer #1: Yes

Reviewer #2: No

4. Is the manuscript presented in an intelligible fashion and written in standard English?

Reviewer #1: No

Reviewer #2: No

Reviewer #1: The article "Prediction of Differentiation Levels in Lung Adenocarcinoma Using Peripheral Blood Inflammatory Cytokines and Tumor Markers" requires a thorough revision, purely from the perspective of presentation. I cannot render much scientific comments until I have clearly understood the manuscript. I urge the authors to kindly consider the following points and resubmit a more 'complete' manuscript. Overall, since the manuscript is submitted to PlosOne and not a specific cancer research journal, I request the authors to keep in mind broad readership (non-clinical) while reformatting their article. Therefore, most acronyms and statistics need to be elaborately described.

1) Please decrypt acronyms in abstract.

2) All tables presentation need to be improved. Table 2, 3 at the moment do not have borders and it is difficult to read. Please consider organizing and presenting better.

3) Table 2: CYFRA21/SCCA TPA abbreviations? P25/P50 meaning?

4) Table 3: High group N=44 and Low group N=159. This difference is nearly 4-folds. I am not exactly certain how the statistics problem will be solved by this difference. The statistics is skewed and compelling the findings become dubious to interpret. At the moment Figure 3,4 is not helping either to understand weightage.

5) None of the figures have legends. Figure 2-4 are incomprehensible at the moment.

Reviewer #2: The authors set out to investigate the predictive value of a combination of peripheral blood inflammatory markers and tumour markers to determine the differentiation levels of lung adenocarcinoma. The peripheral blood biomarkers included neutrophils, monocytes, platelets, and age. The hypothesis stated that certain marker combinations could more accurately reflect the biological characteristics of lung adenocarcinoma. The authors developed a predictive model with an AUC of 0.795, which led them to claim that the peripheral blood biomarkers (including neutrophils, monocytes, platelets, AISI), age, and tumor biomarkers (CEA and ProGRP) are associated with pathological differentiation in lung adenocarcinoma. The claims are significant with respect to advances in biomarker-driven oncology. The model is non-invasive and cost-effective, and may serve as potential predictors for the differentiation of lung adenocarcinoma and may guide personalized treatment planning.

The authors have placed the study in context with literature. However, the authors should take care to define all abbreviations used in the article – as this was not done.

The data and analyses fully support the authors claims.

Whilst the authors have a statistics analysis method, more detail can be included here – considering that the entire predictive model is based on this methodology. For instance, the authors could elaborate on the methodology for generating the decision curve – Figure 6. The same applies to the AUC of the nomogram model (Figure 4) and the calibration curve (Figure 5). Furthermore, the heading “Definition of low and high differentiation” explains the differences between low- and high-differentiated lung adenocarcinoma but includes no methodology. It would have been more thorough for the authors to show cell images representing the differences taken from patient samples, and the authors could consider including this in the manuscript.

All figures are poor in quality - the resolution of each figure needs to be improved. Furthermore, figures have not been presented with figure legends.

The authors should certainly be encouraged to resubmit a revised version as this article could further the current understanding of predictors for the differentiation of lung adenocarcinoma.

Whilst the manuscript is written clearly, there are numerous editorial errors which need to be corrected. Furthermore, the authors should take care to define all abbreviations such that the information is accessible to non-specialists. Furthermore, an introductory sentence or two needs to be added to the abstract.

**Do you want your identity to be public for this peer review?** For information about this choice, including consent withdrawal, please see our Privacy Policy

Reviewer #1: **Yes: ** Joydeep Chakraborty

Reviewer #2: No

---

## [Author Response · Author response to Decision Letter 1]

6 Sep 2025

Dear Editors and Reviewers:

Thank you for your letter and the reviewers’ comments on our manuscript titled “Prediction of Differentiation Levels in Lung Adenocarcinoma Using Peripheral Blood Inflammatory Cytokines and Tumor Markers” (ID: PONE-D-25-11695). We greatly appreciate the valuable feedback, which has significantly contributed to improving our manuscript and providing essential guidance for our research. We have carefully considered the comments and made the necessary revisions, which we hope will meet your approval. The revised sections are highlighted in red in the manuscript. Below are the main corrections and our responses to the reviewers’ comments:

Reviewer 1:

Comment1: Please decrypt acronyms in abstract.

Response Thank you for your valuable feedback.We have introduced the acronyms in the abstract and highlighted them in red.Thank you once again. Your valuable insights have significantly contributed to refining and uplifting the quality of our paper.

Comment2:

All tables presentation need to be improved. Table 2, 3 at the moment do not have borders and it is difficult to read. Please consider organizing and presenting better.

Response

Thank you for pointing this out.In the new manuscript, readers can have a better understanding of the tables in this study. We have added corresponding borders to the tables and beautified the table format.Your insights have been immensely valuable in shaping and enhancing the overall quality of our paper.

Comment3:

Table 2: CYFRA21/SCCA TPA abbreviations? P25/P50 meaning?

Response:

Thank you for the question.P25 and P50, as key percentile statistics, play an irreplaceable role in the research. P25 provides a benchmark for the low-value range, risk threshold setting and outlier identification by marking the lower quartiles of the data distribution. While P50, as the median, accurately reflects the central trend of the data, especially being more representative in the skewed distribution. The combined use of the two can comprehensively depict the distribution characteristics of the data (such as lower limits, centers and skewness), providing an important basis for risk assessment, inter-group comparison and outlier detection. This study introduces these two indicators, aiming to reveal the inherent laws of the data more accurately and lay a solid foundation for subsequent analysis.Your insights have been immensely valuable in shaping and enhancing the overall quality of our paper.

Comment4:

Table 3: High group N=44 and Low group N=159. This difference is nearly 4-folds. I am not exactly certain how the statistics problem will be solved by this difference. The statistics is skewed and compelling the findings become dubious to interpret. At the moment Figure 3,4 is not helping either to understand weightage.

Response

Thank you for your valuable feedback.The reasons for this imbalance may be multifaceted. Firstly, from the patients' perspective, high-differentiation patients may perceive their condition as relatively mild with unobvious symptoms, leading to a less urgent need for medical attention. Additionally, economic factors could play a significant role. Some patients, particularly those with limited financial resources, may hesitate to seek medical care or delay their visits due to concerns about treatment costs.Furthermore, psychological factors should not be overlooked. High-differentiation patients may hold a more optimistic view of their disease prognosis and, consequently, lack the motivation to actively seek medical help.

Meanwhile In our study, we utilized a binary logistic regression model for analysis. When the sample size exceeds 200, the bias in parameter estimation can be considered negligible and therefore does not need to be taken into account[1].

In addition due to the nature of this study as a real-world investigation, the data was collected from a single center. Within the limited timeframe, the number of cases of undifferentiated or moderately differentiated adenocarcinoma was too restricted to allow for robust statistical analysis. Focusing on these two groups can enable a more precise research focus, enhance the accuracy of the study, and generate data that is more relevant to clinical decision-making.Thank you once again. Your valuable insights have significantly contributed to refining and uplifting the quality of our paper.

Reference: 1.Chen, B. (2012). "Medical multivariate analysis design sample size estimation—A comprehensive estimation method for multivariate analysis design sample size." Shanghai Medical Journal (Electronic Edition), 1.04, 58-60.(In chinese)

Comment5:

None of the figures have legends. Figure 2-4 are incomprehensible at the moment.

Response

Thank you for pointing this out.At the end of the manuscript, we have added legends to Figures 2-6 to facilitate readers' better reading and understanding.Your insights have been immensely valuable in shaping and enhancing the overall quality of our paper.

Reviewer 2:

Comment1:

However, the authors should take care to define all abbreviations used in the article as this was not done.

Response:

Thank you for your valuable comment.In the new manuscript, we reinterpreted all the abbreviations.Your feedback has been incredibly beneficial in enhancing the quality of our paper.

Comment2:

Whilst the authors have a statistics analysis method, more detail can be included here considering that the entire predictive model is based on this methodology. For instance, the authors could elaborate on the methodology for generating the decision curve Figure 6. The same applies to the AUC of the nomogram model (Figure 4) and the calibration curve (Figure 5).

Response:

Thank you for your valuable comment.The methods for generating relevant curves and Nomogram models are mentioned in the statistical methods of this paper. Meanwhile, at the end of the new manuscript, we further explained the pictures in the study.Thank you again for your valuable feedback and for helping us to improve the quality and accuracy of our study.

Comment3:

Response:

Thank you for your valuable comment.To systematically clarify the differences between Low Differentiated and High Differentiated lung adenocarcinoma, the following methods were adopted in this study:Sample selection and grouping criteria.

Sample source: Pathologically confirmed cases of lung adenocarcinoma (n=203) collected from the Second Affiliated Hospital of Chongqing Medical University, with a time span of [June 1, 2022 to July 30, 2024].

Grouping basis: According to the 2021 classification criteria for lung tumors of the World Health Organization (WHO), two senior pathologists independently reviewed the pathological sections. Based on characteristics such as the proportion of acini/papilla/micropapilla/solid type and nuclear atypia, the samples were divided into the LD group (Low Differentiated) and the HD group (High differentiated).Thank you for your understanding and valuable feedback.Due to the limitations of width and height, we have improved the quality of the pictures as much as possible as required by the magazine. Currently, there are clearer PDF versions of pictures 2 to 6, which have now been uploaded to the supplementary materials section.

Comment4:

All figures are poor in quality - the resolution of each figure needs to be improved. Furthermore, figures have not been presented with figure legends.

Response:

Due to the limitations of width and height, we have improved the quality of the pictures as much as possible as required by the magazine. Currently, there are clearer PDF versions of pictures 2 to 6, which have now been uploaded to the supplementary materials section.At the end of the manuscript, we have added legends to Figures 2-6 to facilitate readers' better reading and understanding.Thank you again for your valuable feedback and for helping us to improve the quality and accuracy of our study.

Comment5:

Furthermore, an introductory sentence or two needs to be added to the abstract.

Response:

Thank you for your comment.We have added appropriate introduction sentences in the abstract to facilitate readers' better reading and understanding.Thank you once again for your valuable feedback.

---

## [Decision Letter · Decision Letter 1]

12 Oct 2025

Dear Dr. JIang,

Thank you for submitting your manuscript to PLOS ONE. After careful consideration, we feel that it has merit but does not fully meet PLOS ONE’s publication criteria as it currently stands. Therefore, we invite you to submit a revised version of the manuscript that addresses the points raised during the review process.

We look forward to receiving your revised manuscript.

Kind regards,

Zhiling Yu

Academic Editor

PLOS ONE

Journal Requirements:

Reviewers' comments:

Reviewer's Responses to Questions

**Comments to the Author**

Reviewer #1: All comments have been addressed

2. Is the manuscript technically sound, and do the data support the conclusions?

Reviewer #1: Yes

3. Has the statistical analysis been performed appropriately and rigorously?

Reviewer #1: Yes

4. Have the authors made all data underlying the findings in their manuscript fully available?

Reviewer #1: Yes

5. Is the manuscript presented in an intelligible fashion and written in standard English?

Reviewer #1: Yes

Reviewer #1: The article titled "Prediction of Differentiation Levels in Lung Adenocarcinoma Using Peripheral Blood Inflammatory Cytokines and Tumor Markers" certainly is novel and the connecting findings are clinically relevant with abundance of upcoming literature in this area. I have read the manuscript and the response to previous reviewer's comments. I do believe addition of the ROC/AUC, decision curve analysis certainly improved from the previous version. I have a few comments that I urge the authors to consider for another revision.

1) The sensitivity analysis and potential of overfitting risks must be thoroughly explained in the discussion. Overall the limitations section need to be more thorough for this study. If possible please expand on external validation studies, potential geographic area bias. A few future directions on multi-center studies and tumor microenvironment profiling/single cell studies etc. will address these limitations.

2) For the CEA please explain how borderline p values were handled to avoid misinterpretations, Page 14 "The original p-value for CEA was 0.0497.....".

3) I think funding for this study is mentioned twice. Please check acknowledgements and funding sections.

4) Figure resolutions need to enhance.

5) The abstract should report confidence intervals(95%)for AUC And clarifying the relevance of ProGRP (for the revised manuscript) for the context of LUAD.

6) Waiving of informed consent must be clearly declared in methods. Proofread throughout for textual errors.

7) Please consider providing more detailed how decision curve analysis and calibration curves were generated in R/bootstrap approach.

**Do you want your identity to be public for this peer review?** For information about this choice, including consent withdrawal, please see our Privacy Policy

Reviewer #1: **Yes: ** Joydeep Chakraborty

---

## [Author Response · Author response to Decision Letter 2]

20 Nov 2025

List of Responses

Dear Editor and Reviewers:

Thank you for your letter and for the reviewers’ comments concerning our manuscript entitled “Prediction of Differentiation Levels in Lung Adenocarcinoma Using Peripheral Blood Inflammatory Cytokines and Tumor Markers” (ID: PONE-D-25-11695R1). Those comments are all valuable and very helpful for revising and improving our paper, as well as the important guiding significance to our researches. We have studied comments carefully and have made correction which we hope meet with approval. Revised portion are marked in red in the paper. The main corrections in the paper and the responds to the reviewer’s comments are as flowing:

Responds to the comments of editor and reviewers:

Reviewer #1:

1.Comment: If the authors have adequately addressed your comments raised in a previous round of review and you feel that this manuscript is now acceptable for publication, you may indicate that here to bypass the “Comments to the Author” section, enter your conflict of interest statement in the “Confidential to Editor” section, and submit your "Accept" recommendation. All comments have been addressed

Response: Thank you very much for your recognition of this manuscript.

2.Comment: Is the manuscript technically sound, and do the data support the conclusions?The manuscript must describe a technically sound piece of scientific research with data that supports the conclusions. Experiments must have been conducted rigorously, with appropriate controls, replication, and sample sizes. The conclusions must be drawn appropriately based on the data presented. Yes

Response: Thank you very much for your recognition of this manuscript.

3. Comment: Has the statistical analysis been performed appropriately and rigorously? Yes

Response: Thank you very much for your recognition of this manuscript.

4.Comment: Have the authors made all data underlying the findings in their manuscript fully available? The PLOS Data policy requires authors to make all data underlying the findings described in their manuscript fully available without restriction, with rare exception (please refer to the Data Availability Statement in the manuscript PDF file). The data should be provided as part of the manuscript or its supporting information, or deposited to a public repository. For example, in addition to summary statistics, the data points behind means, medians and variance measures should be available. If there are restrictions on publicly sharing data—e.g. participant privacy or use of data from a third party—those must be specified. Yes

Response: Thank you very much for your recognition of this manuscript.

5.Comment: Is the manuscript presented in an intelligible fashion and written in standard English?PLOS ONE does not copyedit accepted manuscripts, so the language in submitted articles must be clear, correct, and unambiguous. Any typographical or grammatical errors should be corrected at revision, so please note any specific errors here. Yes

Response: Thank you very much for your recognition of this manuscript.

6.Comment: Review Comments to the Author. Please use the space provided to explain your answers to the questions above. You may also include additional comments for the author, including concerns about dual publication, research ethics, or publication ethics. (Please upload your review as an attachment if it exceeds 20,000 characters).

Reviewer #1: The article titled "Prediction of Differentiation Levels in Lung Adenocarcinoma Using Peripheral Blood Inflammatory Cytokines and Tumor Markers" certainly is novel and the connecting findings are clinically relevant with abundance of upcoming literature in this area. I have read the manuscript and the response to previous reviewer's comments. I do believe addition of the ROC/AUC, decision curve analysis certainly improved from the previous version. I have a few comments that I urge the authors to consider for another revision.

1)The sensitivity analysis and potential of overfitting risks must be thoroughly explained in the discussion. Overall the limitations section need to be more thorough for this study. If possible please expand on external validation studies, potential geographic area bias. A few future directions on multi-center studies and tumor microenvironment profiling/single cell studies etc. will address these limitations.

Response: Thank you very much for your valuable and insightful comments. We have substantively revised and expanded the Discussion section of our manuscript based on your suggestions. To assess the robustness of the model, we performed Bootstrap resampling to evaluate model performance. The results showed an absolute error of 0.012 between the simulated curve and the actual curve, indicating a high degree of similarity and consistency in the trends of the two curves. Furthermore, the model achieved an AUC of 0.795 (95% CI: 0.726–0.864), which strengthens our confidence in its reliability. Additionally, to mitigate the risk of overfitting, we have supplemented the analysis with multicollinearity test results (Supplementary Table 3), which demonstrated no collinearity among the variables. However, limitations must be acknowledged. First, although multicollinearity tests were employed during model construction to reduce the risk of overfitting, this risk remains in studies with limited sample sizes. Second, the retrospective data used in this study were solely sourced from the Second Affiliated Hospital of Chongqing Medical University, which may introduce potential geographic and population biases. We acknowledged that the model developed from this single center dataset may have limited generalizability, and its performance could decline when applied to populations from other regions, with different ethnic backgrounds, or under varying healthcare settings. Third, we recognized that a truly robust model requires validation with data from other independent centers, but the step had not yet been accomplished in this study. To directly address the limitations identified in this study, we propose several key directions for future research. First, multi center collaborations should be initiated to conduct rigorous external validation of our current model using independent datasets from various institutions. This is an essential step for confirming the model's clinical utility and generalizability. Second, large scale prospective cohort studies are warranted to elucidate the temporal relationship between AISI and the incidence and progression of lung adenocarcinoma. Third, integrating more precise diagnostic tools for lung adenocarcinoma differentiation could enhance the accuracy and reliability of research findings. Furthermore, developing predictive models that incorporate AISI with other biomarkers would help improve prognostic accuracy. Finally, fundamental experimental research focusing on areas such as tumor microenvironment analysis and single-cell studies should be pursued to progressively address the current limitations of the research.

2)For the CEA please explain how borderline p values were handled to avoid misinterpretations, Page 14 "The original p-value for CEA was 0.0497.....".

Response: We thank the reviewer for this insightful and important comment regarding the interpretation of borderline p-values of CEA. We fully agree that p-values close to the conventional alpha level of 0.05 require careful handling to prevent overinterpretation due to random chance. The p-value for CEA was 0.0497, indicating statistical significance at the 0.05 level. Therefore, we have chosen to retain the value of 0.0497, rather than 0.05 or <0.05. This approach effectively prevents potential misinterpretation while clearly informing readers of the statistical rationale for including CEA in the final model.

3)I think funding for this study is mentioned twice. Please check acknowledgements and funding sections.

Response: Thank you for your valuable feedback. We have revised in the revised manuscript. Acknowledgments: We thank the Second Affiliated Hospital of Chongqing Medical University for their support. Funding: This work was supported by Application of single-cell biomolecular analysis in the pathogenesis of lung cancer(cstc2022yejh-bgzxm0051).

4)Figure resolutions need to enhance.

Response: Thank you for your valuable feedback. The figures have been remade and saved in TIFF format (Figure1 - Figure6).

5)The abstract should report confidence intervals(95%)for AUC And clarifying the relevance of ProGRP (for the revised manuscript) for the context of LUAD.

Response: Thank you for your valuable feedback. We have revised in the revised manuscript. AISI (OR = 1.64, 95% CI = 1.08-2.58, p = 0.024), CEA (OR = 1.02, 95% CI = 1.00-1.04, p = 0.0497), ferritin (OR = 1.01, 95% CI = 1.00-1.01, p = 0.010), and Progastrin Releasing Peptide (ProGRP) (OR = 1.03, 95% CI = 1.00-1.07, p = 0.047) were risk factors of low differentiation LUAD. The model achieved an Area Under Curve(AUC) of 0.795 (95%CI: 0.726-0.864).

6)Waiving of informed consent must be clearly declared in methods. Proofread throughout for textual errors.

Response: Thank you for your valuable feedback. We have revised in the revised manuscript. All personal identifiers were removed prior to researcher access, and this retrospective study was conducted in accordance with the ethical principles of the Declaration of Helsinki and was approved by the Second Affiliated Hospital of Chongqing Medical University Ethics Committee (No.85, 2024), with a waiver of informed consent obtained. Additionally, we have thoroughly proofread the manuscript to correct any spelling errors.

7)Please consider providing more detailed how decision curve analysis and calibration curves were generated in R/bootstrap approach.

Response: Thank you for your valuable feedback. We have revised in the revised manuscript. The calibration curve was plotted using the Bootstrap method with 1,000 repetitions. The dashed diagonal line represents the ideal line, which signifies a perfect prediction. The solid line indicates the actual prediction performance of the model. The closer these two lines are to each other, the better the model's predictive accuracy. The results are presented in Figure 5. The absolute error between the simulated curve and the actual curve is 0.012, demonstrating that the two curves follow a highly similar trend and exhibit strong consistency. The decision curve for the established nomogram was constructed. The clinical value of this model increases as its curve moves away from the "treat all" or "treat none" curves. The threshold at which patients benefit from this model ranges from 30% to 95%. Furthermore, the decision curve analysis depicted in Figure 6 confirms the clinical utility of the model, demonstrating a favorable net benefit rate.

7.Comment: PLOS authors have the option to publish the peer review history of their article (what does this mean?). If published, this will include your full peer review and any attached files. Do you want your identity to be public for this peer review? For information about this choice, including consent withdrawal, please see our Privacy Policy. Yes: Joydeep Chakraborty

Response: Thank you very much for your recognition of this manuscript. Your valuable comments are of great importance for the improvement of this manuscript.

Reviewer #2:

The authors set out to investigate the predictive value of a combination of peripheral blood inflammatory markers and tumour markers to determine the differentiation levels of lung adenocarcinoma. The peripheral blood biomarkers included neutrophils, monocytes, platelets, and age. The hypothesis stated that certain marker combinations could more accurately reflect the biological characteristics of lung adenocarcinoma. The authors developed a predictive model with an AUC of 0.795, which led them to claim that the peripheral blood biomarkers (including neutrophils, monocytes, platelets, AISI), age, and tumor biomarkers (CEA and ProGRP) are associated with pathological differentiation in lung adenocarcinoma. The claims are significant with respect to advances in biomarker-driven oncology. The model is non-invasive and cost-effective, and may serve as potential predictors for the differentiation of lung adenocarcinoma and may guide personalized treatment planning.

The study presents the results of original research, and has not been published elsewhere. The data presented in the manuscript support the conclusions drawn.

The authors have done well to meet the applicable standards for the ethics requirements of the study.

Response: Thank you very much for your recognition of this manuscript.

The authors have placed the study in context with literature. However, the authors should take care to define all abbreviations used in the article – as this was not done. The data and analyses support the authors claims.

Response: Thank you for your valuable feedback. We have revised in the revised manuscript. LUAD: lung adenocarcinoma ; BMI, body mass index; CAD, coronary artery disease; AISI: aggregate index of systemic inflammation; CEA: Carcinoembryonic Antigen; CYFRA21-1: Cytokeratin 19 fragment antigen 21-1; NSE: Neuron Specific Enolase; SCCA: Squamous Cell Carcinoma Antigen; TPA: Tissue Polypeptide Antigen; ProGRP: Progastrin Releasing Peptide.

Whilst the authors have a statistics analysis method, more detail can be included here – considering that the entire predictive model is based on this methodology. For instance, the authors could elaborate on the methodology for generating the decision curve – Figure 6. The same applies to the AUC of the nomogram model (Figure 4) and the calibration curve (Figure 5). Furthermore, the heading “Definition of low and high differentiation” explains the differences between low- and high-differentiated lung adenocarcinoma but includes no methodology. It would have been more thorough for the authors to show cell images representing the differences

taken from patient samples, and the authors could consider including this in the manuscript.

Response: Thank you for your valuable feedback. We have revised in the revised manuscript. The calibration curve was plotted using the Bootstrap method with 1,000 repetitions. The dashed diagonal line represents the ideal line, which signifies a perfect prediction. The solid line indicates the actual prediction performance of the model. The closer these two lines are to each other, the better the model's predictive accuracy. The results are presented in Figure 5. The absolute error between the simulated curve and the actual curve is 0.012, demonstrating that the two curves follow a highly similar trend and exhibit strong consistency. The decision curve for the established nomogram was constructed. The clinical value of this model increases as its curve moves away from the "treat all" or "treat none" curves. The threshold at which patients benefit from this model ranges from 30% to 95%. Furthermore, the decision curve analysis depicted in Figure 6 confirms the clinical utility of the model, demonstrating a favorable net benefit rate. Low differentiated LUAD refers to cancer cells with minimal differentiation, showing marked deviations from the epithelial cells of normal lung glands. These cells exhibit irregular shapes, vary in size, have enlarged and darkly stained nuclei, and feature an increased nucleus-to-cytoplasm ratio, indicating a loss of normal cellular characteristics (Figure SIa). High differentiated LUAD consists of cancer cells that are highly differentiated, closely resembling normal lung gland epithelial cells in morphology. These cells appear regular, with smaller nuclei and a normal nucleus-to-cytoplasm ratio(Figure SIb) .

All figures are poor in quality - the resolution of each figure needs to be improved. Furthermore, figures have not been presented with figure legends.

Response: Thank you for your valuable feedback. We have revised in the revised manuscript. The figures have been remade and saved in TIFF format. Figure legends were shown in revised manuscript.

Whilst the manuscript is written clearly, numerous editorial errors need to be corrected. Furthermore,

---

## [Decision Letter · Decision Letter 2]

7 Dec 2025

Prediction of Differentiation Levels in Lung Adenocarcinoma Using Peripheral Blood Inflammatory Cytokines and Tumor Markers

PONE-D-25-11695R2

Dear Dr. Jiang,

We’re pleased to inform you that your manuscript has been judged scientifically suitable for publication and will be formally accepted for publication once it meets all outstanding technical requirements.

Kind regards,

Zhiling Yu

Academic Editor

PLOS One

Additional Editor Comments (optional):

Reviewers' comments:

Reviewer's Responses to Questions

**Comments to the Author**

Reviewer #1: (No Response)

2. Is the manuscript technically sound, and do the data support the conclusions?

Reviewer #1: Yes

3. Has the statistical analysis been performed appropriately and rigorously?

Reviewer #1: Yes

4. Have the authors made all data underlying the findings in their manuscript fully available?

Reviewer #1: Yes

5. Is the manuscript presented in an intelligible fashion and written in standard English?

Reviewer #1: Yes

Reviewer #1: (No Response)

**Do you want your identity to be public for this peer review?** For information about this choice, including consent withdrawal, please see our Privacy Policy

Reviewer #1: **Yes: ** Joydeep Chakraborty

---

## [Editor Report · Acceptance letter]

PONE-D-25-11695R2

PLOS One

Dear Dr. Jiang,

I'm pleased to inform you that your manuscript has been deemed suitable for publication in PLOS One. Congratulations! Your manuscript is now being handed over to our production team.

Kind regards,

on behalf of

Dr. Zhiling Yu

Academic Editor

PLOS One